# The effectiveness of care bundles for reducing caesarean section safely: A systematic review and meta-analysis

Kate O'Doherty[1], Aileen Rothwell[2], Valerie Smith [3]*

1 Royal College of Surgeons in Ireland, Dublin, Ireland, 2 Public and Patient Involvement (PPI) Contributor, Dublin, Ireland, 3 Nursing, Midwifery and Health Systems, University College Dublin, Dublin, Ireland

* valerie.smith@ucd.ie; @valeriesmithUCD

## Abstract

Care bundles, which consist of three or more interventions implemented together, may help address the global rise in caesarean births. The aim of this systematic review is to evaluate the effectiveness of care bundles designed to reduce caesarean section (CS). MEDLINE, CINAHL, CENTRAL and Embase were searched from January 2000 to June 2024 using terms related to CS and care bundles. Grey literature and professional body websites were also searched. Randomised or non-randomised studies reporting on pregnant or labouring women who received a care bundle designed to reduce CS safely were eligible. Data were extracted by two reviewers independently. Pre-specified outcomes included CS (overall, elective, and emergency), assisted vaginal birth, neonatal admission to intensive care, and care bundle compliance. Meta-analyses were undertaken using Review Manager 5.4 and a random effects model. Odds Ratio (OR) with 95% Confidence Interval (CI) were calculated. The certainty of the evidence was assessed using the Grading of Recommendations, Assessment, Development and Evaluation (GRADE) approach. Ten non-randomised studies were included. Care bundles reduced CS overall (OR 0.83, 95% CI 0.72 to 0.97) and emergency CS (OR 0.82, 95% CI 0.68 to 0.98). No differences were observed in assisted vaginal birth (OR 1.10, 95% CI 0.91 to 1.33) or neonatal admission to intensive care (OR 1.06, 95% CI 0.56 to 2.02). Compliance to the care bundles ranged from 50% to 92%. The certainty of the evidence for all outcomes was very low. Randomised trial research is required to better assess care bundle use in reducing CS safely.

## Introduction

Although a life-saving intervention, caesarean section (CS) is not without risk for women and babies especially when performed in the absence of medical need [1,2]. Caesarean section presently accounts for 21% of all births globally, although large

**Data availability statement:** All relevant data are within the manuscript and its Supporting information files, or available on the OSF Project webpage.

**Funding:** Dr K. O'Doherty was part supported by the Health Research Board (Ireland) and the HSC Public Health Agency through Evidence Synthesis Ireland/Cochrane Ireland Fellowship Scheme under Grant number ESI-2021-001. The scheme provides €1,000 for travel expenses related to the review and free access to Evidence Synthesis Ireland training up to 2 years from the start date of the Fellowship. There was no additional external funding received for this study. The funding agency had no role in the design or conduct of the review.

**Competing interests:** The authors have declared that no competing interests exist.

variations in rates exist across countries, for example, from 16.1% in Iceland to 56.9% in Cyprus [3]. The International Federation of Gynaecology and Obstetrics, commenting recently on the alarming increase in CS rates, stated that *'the large variation in CS rates indicates that these rates have virtually nothing to do with evidence-based medicine'* [4]. The International Federation of Gynaecology and Obstetrics concludes by calling on the help of governmental bodies, professional organizations, women's groups, and other stakeholders to implement efforts to reduce unnecessary CS [4].

### Care bundles

Care bundles, introduced in 2001 by the Institute for Healthcare Improvement (IHI) as part of a National Project for Quality Improvement in Healthcare, are defined as a small set of evidence-based interventions for a defined population or patient cohort [5]. The *small set* that makes up a care bundle is generally assumed to be at least three interventions that already exist as part of standard care. When the interventions are implemented together and consistently, the intention is that significantly better outcomes will be achieved. Since 2001, care bundles have been implemented for an array of medical health conditions with demonstrable evidence of effect [6–8].

Care bundles specific to pregnancy, labour, birth and postpartum have also been developed and appear to be gaining momentum. A recent scoping review identified 147 records on the development, implementation and evaluation of care bundles in maternity care for 11 discrete health conditions [9]. These conditions included surgical site infection, obstetric haemorrhage, sepsis, perineal protection, and the safe reduction of CS. Although few randomised trials were identified in the scoping review, many care bundle evaluations were based on before and after studies, rendering syntheses of effectiveness potentially feasible. A recent systematic review, for example, evaluated the use of care bundles for the prevention and treatment of post-partum haemorrhage [10]. Based on 22 randomised and non-randomised studies, the review found evidence of effect with some care bundles, although not with all. Building on the existing evidence and the need for evaluations of care bundles for diverse maternal health conditions [9], the aim of the current review was to evaluate the effectiveness of care bundles designed to reduce CS safely compared to no care bundle use or standard care.

## Materials and methods

The review protocol was prospectively registered with the international prospective register of systematic reviews (PROSPERO) (ID: CRD42024527638). An extended version of the protocol was concurrently registered with Open Science Framework (OSF) (https://osf.io/qvdgn/). The Cochrane Collaboration's methodological standards for effectiveness reviews guided the conduct of the review [11]. Review reporting adhered to the Preferred Reporting of Items in Systematic Reviews and Meta-Analysis (PRISMA) checklist (S1 Table).

### Public and Patient Involvement (PPI)

One member of the review team is a PPI contributor and was involved in the review from protocol development to review completion.

## Inclusion criteria

The population, intervention, comparator, and outcomes (PICO) framework was used to define the review's inclusion criteria. Randomised and non-randomised studies reporting on populations of pregnant or labouring women of any parity or risk status for pregnancy complications were eligible for inclusion. Studies reporting on women with preterm gestation (<37 weeks) or with contraindications for vaginal birth (i.e., placenta previa, placenta accrete) were excluded. The review intervention was a care bundle, meeting the IHI definition of at least three interventions and developed for the purpose of reducing CS, compared to no care bundle or standard care. To be included, the study record had to explicitly describe the intervention as a care bundle.

## Search and selection strategy

To locate eligible studies, the databases of MEDLINE (OVID), CINAHL (EBSCO), the Cochrane Central Register of Controlled Trials (CENTRAL) and Embase (OVID) were searched from January 2000 to June 2024. The 2000 start date was chosen to coincide with the 2001 introduction of the IHI's definition of care bundle. Search terms (S2 Table) were developed by an information retrieval specialist and subjected to Peer Review of Electronic Search Strategies (PRESS) [12] (S3 Table). The search terms were adapted for controlled vocabulary variations across the databases.

The database searches were supplemented by a search of the grey literature websites of OpenGrey System for Information on Grey Literature in Europe, Open University dedicated grey literature site, World Health Organization's International Clinical Trials Registry Platform, and the professional body websites of the Royal College of Obstetricians and Gynaecologists, the American College of Obstetricians and Gynecologists, the Society of Obstetricians and Gynaecologists of Canada, the Royal Australian and New Zealand College of Obstetricians and Gynaecologists, and the Royal College of Midwives. The reference lists of included records were also screened for potentially relevant additional studies not captured in the other searches. Language limitations were not applied to the search strategy, although due to a lack of translation services, only records published in English were included during the screening and selection process. Retrieved records were downloaded to EndNote v20 (Clarivate Analytics, PA, USA), and then uploaded to Covidence (Veritas Health Innovation, Melbourne, Australia) for screening. Screening against the inclusion criteria was undertaken by two reviewers independently, by title and abstract initially, and then by full text. Any differences that arose at each level of screening were resolved with discussion and consensus.

## Quality assessment

Based on findings from the recent scoping review [9], we were aware that studies of diverse designs would likely be included in this review. For this reason, the Effective Public Health Practice Project (EPHPP) quality assessment tool was chosen to appraise the methodological quality of the included studies [13]. The assessment of each included study was undertaken by two reviewers independently and cross-checked for congruency. Any observed differences were resolved through discussion and consensus.

## Data extraction

Data were extracted by two reviewers independently to a pre-designed data extraction form. The form was first piloted on two included studies to assess its utility, with no refinements required. Information extracted included the study reference, country/setting, dates study was conducted, study design, funding source(s), participant characteristics (number, age, parity, gestation, pregnancy risk status), intervention details (care bundle elements) and results related to the reviews' pre-specified outcomes. Data were extracted as reported in the included studies. For any missing or incomplete outcome data, we would have contacted the authors of the study, but this was not required. Once extracted, the reviewers met to corroborate and collate the extracted data in a final data extracted form for each included study. These data extraction forms are available via the OSF project webpage (https://osf.io/qvdgn/).

## Outcome measures

We searched the Core Outcomes in Effectiveness Trials (COMET) database [14] for an established core outcome set (COS) for studies on care bundles, but none was found. We also searched for a COS for studies on CS. One COS project was identified, but this study is ongoing [15]. For this reason, we identified our outcomes based on the review's aim, PPI contributor input, and clinical expertise. The review's primary outcomes were CS overall, elective CS and emergency CS. The review's secondary outcomes were assisted vaginal birth (AVB) (forceps and vacuum combined, forceps only, vacuum only), maternal admission to intensive care, length of hospital stay (in days), maternal postnatal readmission to hospital, maternal satisfaction (as measured in the included studies), Apgar scores <7 at 1 and 5 minutes, admission to neonatal intensive care (NICU), and care bundle compliance (intervention group).

## Data synthesis

Pairwise meta-analyses for each of the review's pre-specified outcomes were undertaken, where feasible, using Review Manager 5.4 software. For dichotomous outcomes, Odds Ratio (OR) with 95% Confidence Interval (CI) were calculated. For continuous outcomes, we planned to report the mean difference (MD) if studies measured the outcome in the same way or the standardized MD if studies measured the same outcome in different ways; however, none of the included studies reported continuous outcome data. Statistical heterogeneity in each meta-analysis was assessed using the I2 statistic. Heterogeneity was considered low if <40%, moderate if 40–60% and substantial if >60% [11]. Anticipating clinical and methodological heterogeneity, all meta-analyses were undertaken using a Random Effects model [11]. If outcome data could not be included in a meta-analysis, for example, insufficient data or data reported in formats not suitable for meta-analysis, the results for these outcomes were reported narratively.

## Assessment of reporting bias

We had planned to investigate potential publication bias using funnel plots and visually assessing funnel plot asymmetry; however, as fewer than 10 studies contributed data to any meta-analysis, funnel plot asymmetry assessment was not performed [11].

## Assessment of heterogeneity

Subgroup analysis was undertaken to explore possible differences between primiparous and multiparous women for the outcome CS overall. During data extraction, we noted differences across the studies in the timing of outcome assessments; a *post hoc* sub-group analysis comparing CS overall measured at ≤ 6-months versus ≥ 6-months post-care bundle implementation was thus conducted.

## Sensitivity analyses

We planned to conduct sensitivity analyses to explore the effect of study quality and study design on CS overall, but these were found unnecessary.

## Certainty of the evidence

We used the Grading of Recommendations, Assessment, Development, and Evaluations (GRADE) approach [16] to assess the certainty of the body of evidence relating to the outcomes for which meta-analyses were feasible. These outcomes were CS overall, emergency CS, AVB (forceps and vacuum combined) and admission to NICU. The GRADE approach uses five considerations, namely, study limitations, consistency of effect, imprecision, indirectness and publication bias [16–20]. We downgraded the evidence for each outcome if concerns presented in any of these considerations.

## Results

### Results of search and selection

The search strategy returned 7580 records of which 7504 were retrieved from database searches and 76 from searches of other sources. Of these, 1208 were identified as duplicates and removed. At title and abstract, 6314 of the remaining 6372 records were screened ineligible and were excluded. The full-text reports of the remaining 58 records were sought of which 57 were retrieved; the full text of one record could not be located. Forty-four of these 57 records were screened ineligible with reasons documented (Fig 1). No non-English language publications were identified in the search and thus none were excluded for this reason. This resulted in the inclusion of 12 records reporting on 10 studies [21–32]. Fig 1 illustrates the search and selection process. The references of all identified studies and those excluded, with reasons, are available in Excel files via the OSF project webpage (https://osf.io/qvdgn/).

### Characteristics of included studies

Table 1 presents the summary characteristics of the included studies. Six of the 10 studies were conducted in the USA, three in Canada, and one was conducted in Australia. All studies used a before and after design. The studies involved single and multiple hospital sites. Six studies included healthy, term nulliparous women, three included multiparous women who had a previous CS and one study included all women. The care bundles evaluated in the studies included different elements although there was overlap of some elements across the care bundles. S4 Table presents all elements of each care bundle.

### Methodological quality of included studies

As three included studies were reported in abstract format only, the methodological quality of seven studies was assessed. None of the studies were deemed high quality overall. Five were assessed as having moderate quality [21–26,32], and two of weak quality [29,31]. Six studies were assessed as moderate on selection bias mostly because participants were somewhat representative, that is restricted to nulliparous women or women with previous CS. Two studies were assessed weak for confounders, mainly due to a lack of information. All studies were assessed weak on blinding. Table 2 presents a summary of the quality assessment for each study. S5 Table presents the complete assessments with explanations.

### Outcome results

None of the included studies measured the pre-specified outcomes of AVB (forceps only; vacuum only), maternal admission to intensive care, length of stay, and maternal postnatal readmission to hospital.

Eight included studies reported CS overall. Data from six of these were included in a meta-analysis, and for two, the data are reported narratively. The meta-analysis demonstrated fewer CS with care bundle use compared to standard care (OR 0.83, 95% CI 0.72 to 0.97) (Fig 2). Heterogeneity, however, was high at 80%.

Exploring heterogeneity through sub-group analyses revealed a slightly greater effect estimate for multiparous women with previous CS (OR 0.73, 95% CI 0.64 to 0.83, $I^2$ 0%) than in nulliparous women (OR 0.83, 95% CI 0.70 to 0.98, $I^2$ 54%), although the test for sub-group differences was not significant (S6 Fig 1.1). Duration of care bundle use also demonstrated differential effects, with fewer CS found at shorter (OR 0.76, 95% CI 0.59 to 0.98, $I^2$ 63%) than longer (OR 0.94, 95% CI 0.74 to 1.19, $I^2$ 68%) durations post care bundle implementation although the test for sub-groups differences was not significant (S6 Fig 1.2)

In the two studies that provided narrative data, reductions in CS overall in favour of care bundles were also noted; from 28.5% to 26.9% (p = 0.011) in nulliparous women [22], 36.1% to 31.3% (p < 0.001) in women who were induced [22], and from 22% to 21% over an eight-week period [31].

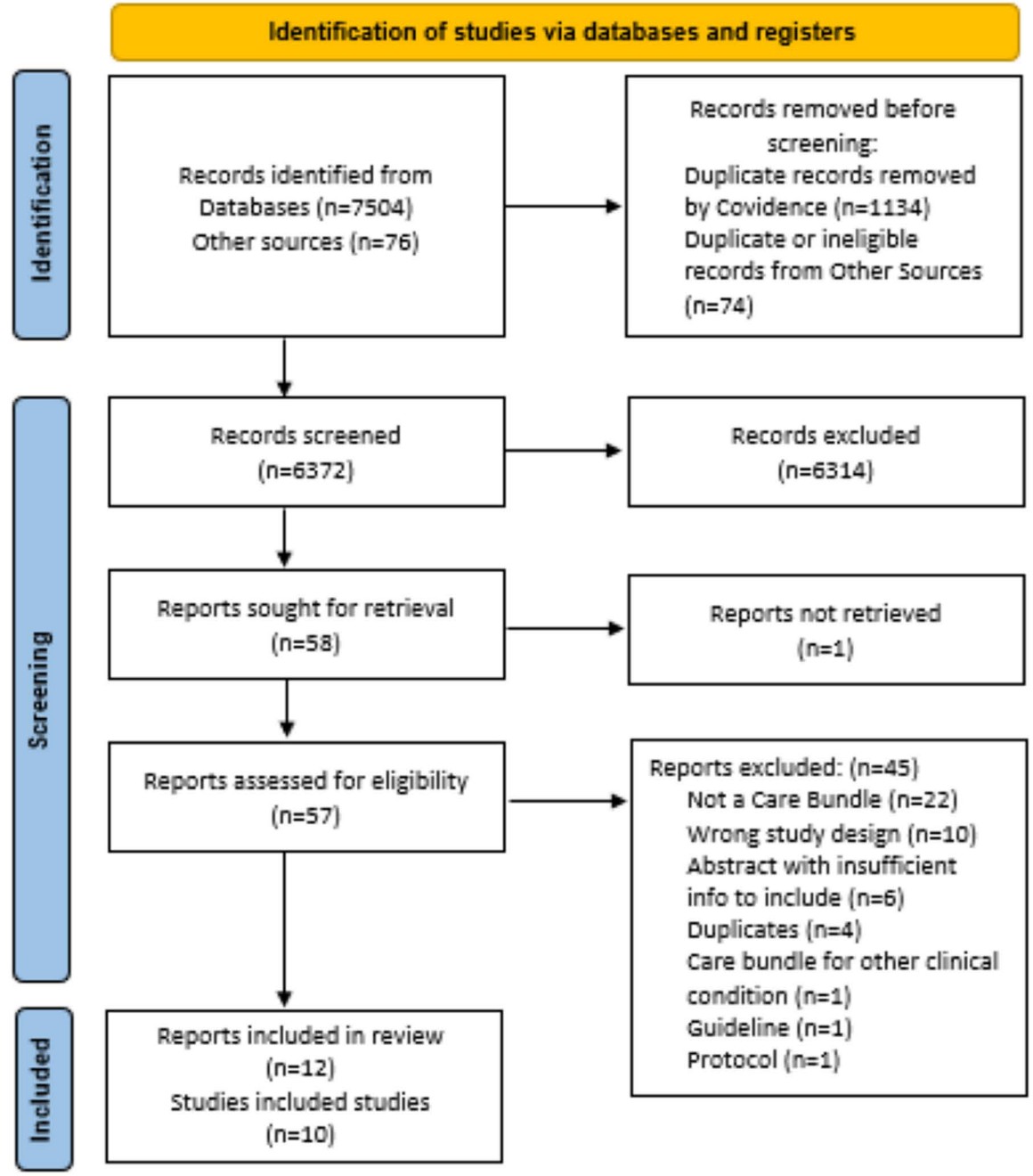

**Fig 1. Search and selection flow diagram.**

Two studies reported narratively on elective repeat CS. Fan reported reductions in CS rates of 10.3% and 14.9% at two study sites [24]. Miazga reported a similar decrease from 63% pre-bundle to 52% 12-months post-bundle implementation [26]. Two studies also reported on emergency CS [26,30]. The meta-analysis demonstrated fewer emergency CS in favour of care bundles (OR 0.82, 95% CI 0.68 to 0.98, $I^2$ 0%) (S6 Fig 2)

**Table 1. Summary characteristics of included studies.**

| Study ID | Aim | Country & setting | Study design & Dates | Study dates | Participants | Data collection |
|---|---|---|---|---|---|---|
| Bell 2017[21] | To implement a systematic approach for care of nulliparous women to reduce CS rates at three hospitals | USA Three acute care hospitals as part of the Carolinas Health Care System | Before-After Retrospective | Jan 2015 to Jun 2015 (before) Jul 2016 to Dec 2016 (after) | Nulliparous women with a term, singleton pregnancy in a vertex presentation (n = 434 before; n = 401 after) | Medical records |
| Callaghan-Koru 2021a[22], 2021b[23] (abstract) | To evaluate the effects of Maryland's collaborative on state-level nulliparous, term, singleton, vertex CS rates | USA: 31 birthing hospitals in Maryland | Before-After Prospective | Jun 2016 to Nov 2018 (12 months before; 30 months after) | Nulliparous women with term, singleton, vertex fetuses (n = unclear) | Survey and other sources (e.g., hospital profiles) |
| Fan 2021[24] (abstract) | To decrease ERCS rates in women eligible for TOLAC by 5% over 1 year through implementation of a TOLAC bundle. | Canada: Credit Valley Hospital and Mississauga Hospital | Before-After Retrospective | No dates but notes 1-year before and 15-months after | Women with previous CS (n = not provided) | Medical records |
| Garpiel 2018[25] | To implement second-stage practice bundle to promote safe outcomes including method of birth and women's birth experience | USA: 34 hospitals | Before-After Retrospective | No dates but notes 2 months before; 4-months after | Low-risk nulliparous women in the second stage of labour (n = 6012 before; n = 11223 after) | Medical records |
| Miazga 2020[26] | To decrease the CS rate in Robson 5 patients eligible for TOLAC with a multifaceted TOLAC bundle | Canada: St. Michael's Hospital, an inner-city tertiary care centre with approx. 2800 births per year | Before-After Retrospective | Feb 2017 to Jan 2018 (before) June 2018 to May 2019 (after) | Women eligible for TOLAC; pregnancy with a cephalic presentation and only one previous CS (n = 247 before; n = 214 after) | Medical records |
| Miazga 2021a[27] (abstract); 2021b[28] (abstract) | To study the efficacy of a TOLAC bundle in decreasing CS rates across five hospitals. | Canada: Five hospitals (two tertiary-care and three community hospitals) | Before-After Retrospective | Dates not provided | Women eligible for TOLAC (n = 2244 before; n = 2096 after) | Medical records |
| Page 2021[29] | To reduce the rate of primary CS among women with low-risk pregnancies without a prior birth by implementing the Promoting Comfort in Labor safety bundle | USA: A Level II regional hospital in Virginia | Before-After (QIP) Prospective | Jan 2015 to Mar 2016 (before) Sept 2017 to June 2019 (after) | Low risk nulliparous pregnancies (n = 864 before; n = 869 after) | Medical records |
| Ryan 2012[30] (abstract) | To improve the overall care and birth outcomes for primiparous women attending the Royal Women's Hospital | Australia: Royal Women's Hospital, Melbourne | Before-After Prospective | 1-year pre-Oct 2010 (before) 1-year post-Oct 2010 (after) | Normal, healthy primiparous labouring women (n = 1286 before; n = 1861 after) | Medical records |
| Telfer 2021[31] | To implement an evidence-based labour care bundle to reduce early labour admissions and improve adherence to labour guidelines shown to reduce CS | USA: Birthing unit in an affiliated teaching hospital in the Northeast | Before and After QIP with four Plan Do Study Act cycles over 8 weeks | Dates not provided | 68 childbearing NTSV families who participated in the bundle | Triage log, delivery log and individual chart reviews |
| Tolcher 2016[32] | To evaluate the impact of an intrapartum care bundle on adverse obstetrics outcomes | USA: Mayo Clinic Rochester, a tertiary care academic centre | Before-After Retrospective | Jan 2011 to Feb 2012 (before) July 2012 to Dec 2013 (after) | All women who gave birth at the Mayo Clinic during the study period (n = 2349 before; n = 2856 after) | Electronic medical record and diagnosis codes |

CS: caesarean section, ERCS: elective repeat caesarean section, TOLAC: trial of labour after caesarean.

**Table 2.  Summary EPHPP Results.**

| Study ID | Selection Bias | Study Design | Confounding | Blinding | Data Collection | Withdrawals | Intervention Integrity | Global Rating |
|---|---|---|---|---|---|---|---|---|
| Bell 2017[21] | M | M | S | W | M | S | 80-100% | M |
| Callaghan-Koru 2021a[22] | M | M | M | W | M | S | 80-100% | M |
| Garpiel 2018[25] | M | M | M | W | M | S | 80-100% | M |
| Miazga 2020[26] | M | M | M | W | M | S | 80-100% | M |
| Page 2021[29] | M | M | W | W | M | S | 80-100% | W |
| Telfer 2021[31] | M | M | W | W | M | S | < 60% | W |
| Tolcher 2016[32] | S | M | S | W | M | S | Can't tell | M |

S: Strong, M: Moderate, W: Weak.

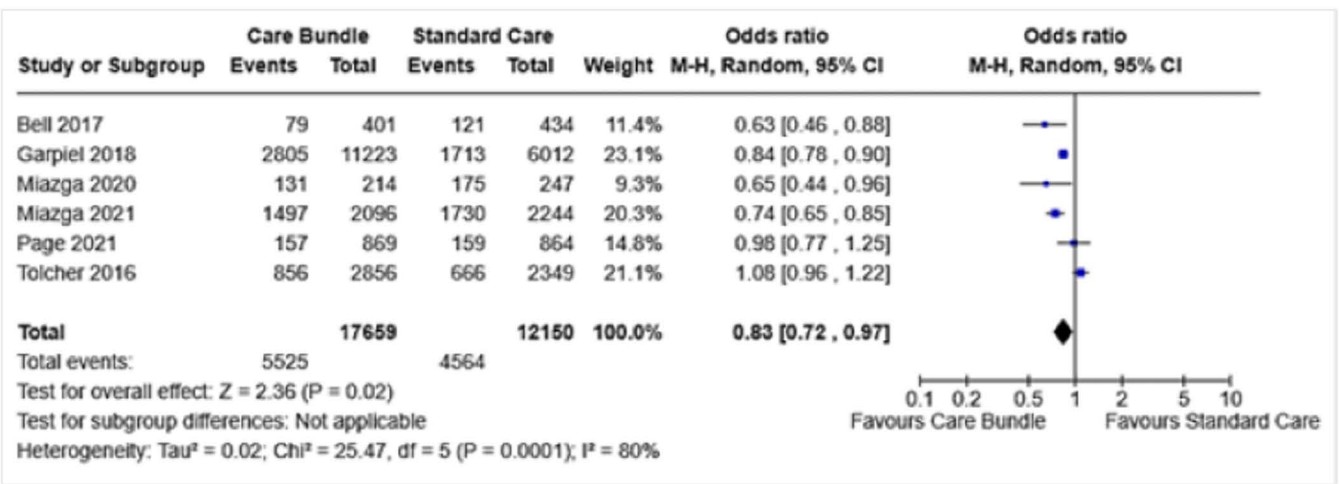

**Fig 2.  CS overall.**

Four studies reported on the outcome of AVB forceps and vacuum birth combined. Although AVB was higher in the care bundle group, the difference was not significant (OR 1.10, 95% CI 0.91 to 1.33, I² 68%) (Fig 3).

One study reported maternal satisfaction with all participants reporting their overall experience as excellent [31]. Two studies reported on Apgar scores <7 at five minutes after birth. Fan narratively reported no differences between the groups [24]. Bell also found no significant differences between the groups (OR 0.86, 95% CI 0.23 to 3.24) [21].

Five studies reported on the outcome of neonatal admission to NICU, of which data from three were combined in a meta-analysis. The overall effect estimate showed no difference between the groups (OR 1.06, 95% CI 0.56 to 2.02, I² 70%) (Fig 4). The remaining two studies also narratively reported no differences between the groups [27,31].

Compliance to the care bundle intervention was reported in three studies. Compliance rates ranged from 50% [22] to 58% [31] and 91.5% [21]. Compliance to individual care bundle elements was also reported in three studies. Telfar described a 47% walking path use and that the partograph and/or pre-CS checklist was applied in 53% of care episodes [32]. Callaghan-Koru reported assessment of fetal heart rate and standardized induction scheduling as the commonest elements implemented (80.6%) with the integration of doulas into the care team the lowest (9.7%) [22]. Garpiel reported significant baseline to 4 months post implementation improvements in all elements of the care bundle (p-values <0.001) other than with the care intervention of removing the indwelling catheter prior to second stage pushing which remained unchanged (73% at baseline and post intervention) [25].

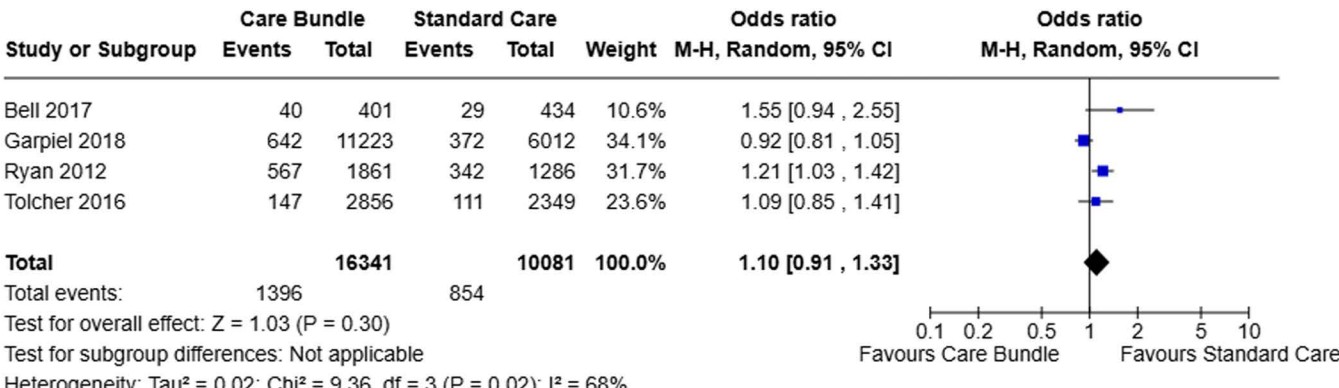

**Fig 3. AVB (forceps and vacuum combined).**

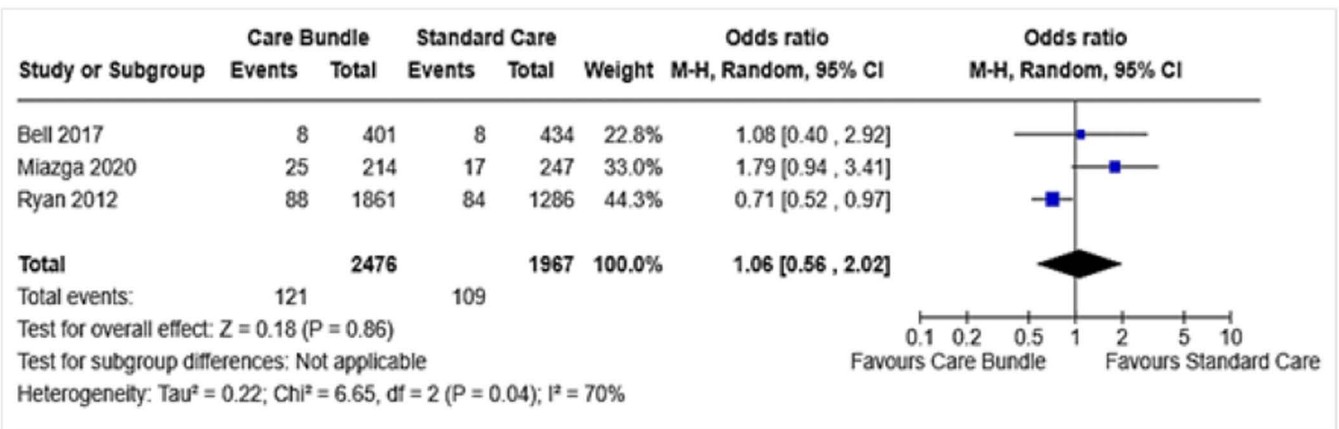

**Fig 4. Admission to NICU.**

## Certainty of the evidence

Table 3 presents the GRADE profile for the four outcomes where it was possible to generate evidence certainty. The evidence for all outcomes was very low indicating that the true effect is likely to be substantially different from the estimate of the effect. Downgrading was mainly due to the non-randomised study designs, methodological limitations, and high heterogeneity.

## Discussion

Use of care bundles are predicated on the notion that evidence-based care practices, when implemented together and consistently, lead to improved health and care outcomes [5]. The findings of this review suggest that care bundles designed to reduce CS safely may reduce CS. No evidence of increased harm for women and babies was identified. The evidence, however, is of very low certainty and fraught with clinical, methodological and statistical heterogeneity. Although eight studies provided data on CS overall, limited data to assess elective and repeat CS were available. Furthermore, four included studies were limited to nulliparous women only and three to populations planning VBAC. Classified as Robson Group 1 and 5 respectively [33], these populations are known to contribute highly to overall CS rates [34]. In this regard,

**Table 3. Summary of Findings Table.**

| Outcome | Relative effect (95% CI) | Anticipated absolute effects (95% CI) | | | Certainty |
|---|---|---|---|---|---|
| | | Standard Care | Care Bundles | Difference | |
| CS Overall (6 studies) | OR 0.83 (0.72 to 0.97) | 37.6% | 33.3% (30.2 to 36.9) | 4.3% fewer | ⊕○○○ Very low a,b,c,d |
| Emergency CS (2 studies) | OR 0.82 (0.68 to 0.98) | 17.2% | 14.5% (12.3 to 16.9) | 2.6% fewer | ⊕○○○ Very low a,c,d |
| AVB (Forceps + Vacuum) (4 studies) | OR 1.10 (0.91 to 1.33) | 8.5% | 9.2% (7.8 to 11) | 0.8% more | ⊕○○○ Very low a,b,c,d |
| NICU admission (3 studies) | OR 1.06 (0.56 to 2.02) | 5.5% | 5.9% (3.2 to 10.6) | 0.3% more | ⊕○○○ Very low a,b,c,d |

Explanations: [a] Most information from studies of moderate quality; [b] Substantial heterogeneity; [c] Clinical differences in populations and interventions; [d] Data collected mostly from medical records (observational).

evaluating these groups discretely has clinical value as they have the potential to benefit most from an intervention that reduces CS rates. Nonetheless, limiting evaluations in this way does reduce the ability to understand the broad applicability and generalisability of care bundles designed to reduce CS overall.

Although the elements of some care bundles in this review overlapped (Supplementary File 4), many care bundles consisted of diverse elements, highlighting a potential issue with how these care bundles are being developed. The inclusion of opinion or non-evidence-based elements in developing a care bundle is also problematic and transgresses the very notion on which care bundles are developed. Although several systematic reviews of interventions for reducing CS, including a comprehensive overview of reviews [35] have been published, these publications focus on single interventions or complex interventions which are not necessarily packaged or implemented as a care bundle. In this regard, this is the first systematic review that has explored the clinical effectiveness of care bundles designed specifically for reducing CS. The translation of evidence from the review, however, is hindered. Contributory factors include clinical and methodological heterogeneity, low-certainty evidence, and variation in compliance to the care bundle elements, with none of the studies that measured this outcome reporting 100% compliance. Although the reasons for non-compliance were not fully explicated in the studies, qualitative studies on care bundles for other maternal health conditions have cited a lack of certainty about the evidence supporting care bundle elements, compromised autonomy, and a lack of training time on implementing the care bundle as reasons for non-compliance to care bundle implementation [36–38].

These findings point to a need for a multinational, collective approach to care bundle development, for example, adapting or using established consensus methodology [39]. In this way, the development of multiple care bundles for the same maternal condition would be avoided and consensus on the minimum elements that should comprise the care bundle could be achieved. Qualitative enquiry, as part of this process and centred on care bundle acceptability and feasibility, could also be attained.

Although elements across the care bundles in this review were diverse, reassuringly, we found that almost all included elements appeared evidenced based, although the strength of the evidence underpinning the elements varied. Audit and feedback or debriefing, for example, was an element in three care bundles. Evidence from a systematic review evaluating the use of the Robson 10-Group classification system for audit of CS found a reduction in CS rates; however, the authors simultaneously highlight that none of the six included studies were randomised or controlled, and all had a high risk of bias [40]. Similarly, for partogram use, which was an element in two care bundles, evidence of effect for reducing CS in low resource settings has been identified [38], but the evidence for partogram use compared to no use for reducing CS in all women undergoing spontaneous labour at term is of very low certainty [41].

## Strengths and limitations

Although robust, well established systematic review methods were used in this review, the data contributing to analyses were from reports of retrospective non-randomised studies. These types of studies provide lower-level hierarchical evidence [42] and are prone to bias [43]. Any causal relationship between care bundles and a reduction in CS cannot, thus, be confidently derived. A further limitation was the identification of studies from high resource settings only, with none identified from low to middle income countries. This limits the generalisability of the review to a non-global maternity population. Lastly, we included reports of English language studies only, which has the potential to introduce language bias. Reassuringly, however, no studies were excluded based on language.

## Conclusion

Caesarean section rates continue to increase worldwide with implications for both mothers and infants. A critical question is how to move forward with measures which can halt this trajectory safely. Care bundles may appear promising, yet the current evidence is of very-low certainty and is fraught with clinical and methodological heterogeneity. A more global, standardised approach to developing care bundles for reducing CS safely, followed by robust randomised trial evaluations that involve feasibility and acceptability as well as clinical and cost effectiveness is required.

## Supporting information

**S1 Table. PRISMA checklist.**
(DOCX)

**S2 Table. Search Strategy and Results.**
(DOCX)

**S3 Table. PRESS checklist completed.**
(DOCX)

**S4 Table. Description of the Care Bundles.**
(DOCX)

**S5 Table. EPHPP Assessments.**
(XLSX)

**S6 Fig. Additional Forest Plots Figs 1.1, 1.2, 2.**
(DOCX)

## Acknowledgments

We are grateful for Dr D. O'Malley for taking time to undertake the PRESS review of our search strategy.

## Author contributions

**Conceptualization:** Valerie Smith.

**Formal analysis:** Kate O'Doherty, Valerie Smith.

**Funding acquisition:** Kate O'Doherty.

**Investigation:** Aileen Rothwell.

**Methodology:** Kate O'Doherty, Aileen Rothwell, Valerie Smith.

**Project administration:** Valerie Smith.

**Supervision:** Valerie Smith.

**Writing – original draft:** Kate O'Doherty, Valerie Smith.

**Writing – review & editing:** Kate O'Doherty, Aileen Rothwell, Valerie Smith.

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
