## [Decision Letter · Decision Letter 0]

Jun 26 2025

PONE-D-25-09274The effectiveness of care bundles for reducing caesarean section safely: A systematic review and meta-analysisPLOS ONE

Dear Dr. Smith,

Thank you for submitting your manuscript to PLOS ONE. After careful consideration, we feel that it has merit but does not fully meet PLOS ONE’s publication criteria as it currently stands. Therefore, we invite you to submit a revised version of the manuscript that addresses the points raised during the review process.

We look forward to receiving your revised manuscript.

Kind regards,

Umberto Simeoni

Academic Editor

PLOS ONE

**Journal Requirements:**

1. When submitting your revision, we need you to address these additional requirements. Please ensure that your manuscript meets PLOS ONE's style requirements, including those for file naming. The PLOS ONE style templates can be found at https://journals.plos.org/plosone/s/file?id=wjVg/PLOSOne_formatting_sample_main_body.pdf and https://journals.plos.org/plosone/s/file?id=wjVg/PLOSOne_formatting_sample_main_body.pdfhttps://journals.plos.org/plosone/s/file?id=ba62/PLOSOne_formatting_sample_title_authors_affiliations.pdf 2. In the case of your submission, we note that you mention that language restrictions were imposed on the literature searches. However this is later contradicted to say that no non-English language literature were excluded due to language restrictions. Please can you update these statements to clarify that no studies were excluded based on language. Please note that a lack of translation services is not an adequate justification for imposing language restrictions. 3. Thank you for stating in your Funding Statement: Dr K. O’Doherty is part supported by the Health Research Board (Ireland) and the HSC Public Health Agency through Evidence Synthesis Ireland/Cochrane Ireland under Grant number ESI-2021-001. The funding agency has no role in the design or conduct of the review. Please provide an amended statement that declares *all* the funding or sources of support (whether external or internal to your organization) received during this study, as detailed online in our guide for authors at http://journals.plos.org/plosone/s/submit-now.  Please also include the statement “There was no additional external funding received for this study.” in your updated Funding Statement. Please include your amended Funding Statement within your cover letter. We will change the online submission form on your behalf. 4. As required by our policy on Data Availability, please ensure your manuscript or supplementary information includes the following:  A numbered table of all studies identified in the literature search, including those that were excluded from the analyses.   For every excluded study, the table should list the reason(s) for exclusion.   If any of the included studies are unpublished, include a link (URL) to the primary source or detailed information about how the content can be accessed.  A table of all data extracted from the primary research sources for the systematic review and/or meta-analysis. The table must include the following information for each study:  Name of data extractors and date of data extraction  Confirmation that the study was eligible to be included in the review.   All data extracted from each study for the reported systematic review and/or meta-analysis that would be needed to replicate your analyses.  If data or supporting information were obtained from another source (e.g. correspondence with the author of the original research article), please provide the source of data and dates on which the data/information were obtained by your research group.  If applicable for your analysis, a table showing the completed risk of bias and quality/certainty assessments for each study or outcome.  Please ensure this is provided for each domain or parameter assessed. For example, if you used the Cochrane risk-of-bias tool for randomized trials, provide answers to each of the signalling questions for each study. If you used GRADE to assess certainty of evidence, provide judgements about each of the quality of evidence factor. This should be provided for each outcome.   An explanation of how missing data were handled.  This information can be included in the main text, supplementary information, or relevant data repository. Please note that providing these underlying data is a requirement for publication in this journal, and if these data are not provided your manuscript might be rejected.   

Reviewers' comments:

Reviewer's Responses to Questions

**Comments to the Author**

1. Is the manuscript technically sound, and do the data support the conclusions?

Reviewer #1: Yes

Reviewer #2: Yes

2. Has the statistical analysis been performed appropriately and rigorously? 

Reviewer #1: Yes

Reviewer #2: Yes

3. Have the authors made all data underlying the findings in their manuscript fully available?

Reviewer #1: Yes

Reviewer #2: Yes

4. Is the manuscript presented in an intelligible fashion and written in standard English?

Reviewer #1: Yes

Reviewer #2: Yes

5. Review Comments to the Author

**Reviewer #1: ** I would like to thank authors firstly to choose such an important topic in obstetrics particularly,nowadys C.S.rate show a steady increase and secondly for giving such effort and time to produce such a comprhensive and informative study.

**Reviewer #2: ** Thank you for asking me to review the manuscript.

It is a well written systematic review. The Abstract, introduction, materials and method, findings and discussion were properly carried out, however I have the following reservations/ comments:

1. The use of acronyms to begin sentences is generally discouraged eg CS and FIGO in lines 36 and 41 respectively.

2. In Materials and methods, things planned but not done because they were unnecessary need not be elaborated.. eg sensitivity analysis and heterogeneity assessment. These could be replaced in just a statement that `a planned sensitivity analysis and heterogeneity assessment were found unnecessary`, in one sentence instead of the details.

3. A few omissions manifesting as grammatical error are highlighted in the annotations.

Thank you for asking for my opinion.

Dr Abah MG

6. PLOS authors have the option to publish the peer review history of their article (what does this mean? ). If published, this will include your full peer review and any attached files.

**Do you want your identity to be public for this peer review?** For information about this choice, including consent withdrawal, please see our Privacy Policy .

Reviewer #1: **Yes: ** Mohsen M A Abdelhafez

Reviewer #2: No

---

## [Author Response · Author response to Decision Letter 1]

19 May 2025

Response to Reviewers

Journal Requirements:

Response: We have now updated the manuscript to meet the Journal Style, including correct formatting of Headings (Levels 1-3), Supp Files (i.e., now labelled S1 Table, S6 Figure, etc.), and intext corrections (tracked)

2. In the case of your submission, we note that you mention that language restrictions were imposed on the literature searches. However, this is later contradicted to say that no non-English language literature was excluded due to language restrictions. Please can you update these statements to clarify that no studies were excluded based on language. Please note that a lack of translation services is not an adequate justification for imposing language restrictions.

Response: Yes, this is correct. We did not impose language restrictions on the search, but restricted during the screening process; however, no non-English publications were identified during screening and thus none were excluded for this reason. We have now clarified in Line 101 of Methods by adding: “…only records published in English were included during the screening and selection process” and in the Results (Line 180-181): “No non-English language publications were identified in the search and thus none were excluded for this reason”

Dr K. O’Doherty is part supported by the Health Research Board (Ireland) and the HSC Public Health Agency through Evidence Synthesis Ireland/Cochrane Ireland under Grant number ESI-2021-001. The funding agency has no role in the design or conduct of the review.

Response: The funding statement is updated in the Cover letter to indicate the amount and purpose of the funding received by K. O’Doherty, and the declaration that ‘There was no additional external funding received for this study’

4. As required by our policy on Data Availability, please ensure your manuscript or supplementary information includes the following:

Response: Excel Files of all retrieved studies (numbered) and those excluded (with reasons) are available via the OSF project webpage. A sentence directing the readers to these has been added to the main manuscript (Lines 182-184) as follows: “The references of all identified studies and those excluded, with reasons, are available in Excel files via the projects OSF webpage (https://osf.io/qvdgn/).”

Response: All data extraction forms for the included studies, including dates data were extracted, by whom and agreed by whom, are now available via the OSF project webpage. A sentence for the reader about this has been added to the main manuscript (Lines 119-123) as follows: “Once extracted, the reviewers met to corroborate and collate the extracted data in a final data extracted form for each included study. These data extraction forms are available via the OSF project webpage (https://osf.io/qvdgn/).”

Response: A detailed quality assessment Table, including the assessments for each Domain in the EPHPP tool, was submitted with the original submission; Supplementary File 5 (now relabelled in the manuscript as S5 Table). The GRADE assessments were completed in GRADEPro with the reasons for downgrading on each evidence factor for each outcome signposted in the footnotes in Table 3 Summary of Findings Table.

An explanation of how missing data were handled. This information can be included in the main text, supplementary information, or relevant data repository. Please note that providing these underlying data is a requirement for publication in this journal, and if these data are not provided your manuscript might be rejected.

Response: A sentence regarding missing data is now added to the Data Extraction section in the Methods (Line 119-120) as follows: “Data were extracted as reported in the included studies. For any missing or incomplete outcome data, we would have contacted the authors of the study, but this was not required.”

5. Please review your reference list to ensure that it is complete and correct. If you have cited papers that have been retracted, please include the rationale for doing so in the manuscript text or remove these references and replace them with relevant current references. Any changes to the reference list should be mentioned in the rebuttal letter that accompanies your revised manuscript. If you need to cite a retracted article, indicate the article’s retracted status in the References list and include a citation and full reference for the retraction notice.

Response: The reference list has been checked and edited to ensure all references confirm to the Journal style. This included expanding to ensure up to six authors are included and inserting DOI’s where these were missing. No retracted papers have been cited.

Reviewers’ Comments to the Author

Reviewer #1:

I would like to thank authors firstly to choose such an important topic in obstetrics particularly, nowadays, CS rates show a steady increase and secondly for giving such effort and time to produce such a comprehensive and informative study.

Response: Thank you for taking the time to review our work, and for your positive feedback.

Reviewer #2:

Thank you for asking me to review the manuscript.

It is a well written systematic review. The Abstract, introduction, materials and method, findings and discussion were properly carried out, however I have the following reservations/ comments:

Response: Thank you for taking the time to review our work, and for your helpful comments. We have attended to these as follows:

1. The use of acronyms to begin sentences is generally discouraged e.g. CS and FIGO in lines 36 and 41 respectively.

Response: We have changed these to their full terms at the start of the sentences as highlighted (now Lines 37 and 42, respectively, as well as Line 319)

2. In Materials and methods, things planned but not done because they were unnecessary need not be elaborated e.g. sensitivity analysis and heterogeneity assessment. These could be replaced in just a statement that `a planned sensitivity analysis and heterogeneity assessment were found unnecessary’, in one sentence instead of the details.

Response: We have reduced the elaborate text, and condensed the information to one sentence (Lines 160-161) as follows: “We planned to conduct sensitivity analyses to explore the effect of study quality and study design on CS overall, but these were found unnecessary.”

3. A few omissions manifesting as grammatical error are highlighted in the annotations.

Thank you for asking for my opinion.

Response: Thank you, we have incorporated all the highlighted edits, and edited Line 267 in the Discussion for clarity

Response: PACE enhanced Figures have been included in the revised version

---

## [Editor Report · Decision Letter 1]

The effectiveness of care bundles for reducing caesarean section safely: A systematic review and meta-analysis

PONE-D-25-09274R1

Dear Dr. Smith,

We’re pleased to inform you that your manuscript has been judged scientifically suitable for publication and will be formally accepted for publication once it meets all outstanding technical requirements.

Kind regards,

Umberto Simeoni

Academic Editor

PLOS ONE
---

## [Editor Report · Acceptance letter]

PONE-D-25-09274R1

PLOS ONE

Dear Dr. Smith,

I'm pleased to inform you that your manuscript has been deemed suitable for publication in PLOS ONE. Congratulations! Your manuscript is now being handed over to our production team.

Kind regards,

on behalf of

Prof. Umberto Simeoni

Academic Editor

PLOS ONE